# N,N-Dimethylglycine Sodium Salt Exerts Marked Anti-Inflammatory Effects in Various Dermatitis Models and Activates Human Epidermal Keratinocytes by Increasing Proliferation, Migration, and Growth Factor Release

**DOI:** 10.3390/ijms241411264

**Published:** 2023-07-09

**Authors:** Alexandra Lendvai, Gabriella Béke, Erika Hollósi, Maike Becker, Jörn Michael Völker, Erik Schulze zur Wiesche, Attila Bácsi, Tamás Bíró, Johanna Mihály

**Affiliations:** 1Department of Immunology, Faculty of Medicine, University of Debrecen, 4032 Debrecen, Hungary; lendvai.alexandra@med.unideb.hu (A.L.); beke.gabriella@med.unideb.hu (G.B.); hollosi.erika@med.unideb.hu (E.H.); etele@med.unideb.hu (A.B.); biro.tamas.pr@gmail.com (T.B.); 2Gyula Petrányi Doctoral School of Clinical Immunology and Allergology, University of Debrecen, 4032 Debrecen, Hungary; 3Dr. Kurt Wolff GmbH & Co. KG, 33611 Bielefeld, Germany; maike.becker@drwolffgroup.com (M.B.); joern-michael.voelker@drwolffgroup.com (J.M.V.); erik-schulze-zur-wiesche@drwolffgroup.com (E.S.z.W.); 4Dr. August Wolff GmbH & Co. KG Arzneimittel, 33611 Bielefeld, Germany; 5ELKH-DE Allergology Research Group, 4032 Debrecen, Hungary

**Keywords:** N,N-dimethylglycine sodium salt, DMG-Na, anti-inflammatory, dermatitis models, human epidermal keratinocytes, HaCaT

## Abstract

N,N-dimethylglycine (DMG) is a naturally occurring compound being widely used as an oral supplement to improve growth and physical performance. Thus far, its effects on human skin have not been described in the literature. For the first time, we show that N,N-dimethylglycine sodium salt (DMG-Na) promoted the proliferation of cultured human epidermal HaCaT keratinocytes. Even at high doses, DMG-Na did not compromise the cellular viability of these cells. In a scratch wound-closure assay, DMG-Na augmented the rate of wound closure, demonstrating that it promotes keratinocyte migration. Further, DMG-Na treatment of the cells resulted in the upregulation of the synthesis and release of specific growth factors. Intriguingly, DMG-Na also exerted robust anti-inflammatory and antioxidant effects, as assessed in three different models of human keratinocytes, mimicking microbial and allergic contact dermatitis as well as psoriasis and UVB irradiation-induced solar dermatitis. These results identify DMG-Na as a highly promising novel active compound to promote epidermal proliferation, regeneration, and repair, and to exert protective functions. Further preclinical and clinical studies are under investigation to prove the seminal impact of topically applied DMG-Na on relevant conditions of the skin and its appendages.

## 1. Introduction

N,N-dimethylglycine (DMG) is part of the endogenous homocysteine pathway and is found in several food sources and plants [1,2], and the sodium salt of N,N-dimethylglycine (DMG-Na) has been widely used as a food supplement in animal husbandry to increase the feed/weight ratio, induce fattening, or boost physical performance [2,3,4,5,6,7,8,9,10,11]. Moreover, DMG is also approved as a safe, non-fuel food supplement for human application [2,12,13]. Besides these effects, it has also been proposed that DMG and its sodium salt may function as a source of glycine for glutathione synthesis, and hence may improve cellular and tissue antioxidant capacities. Indeed, in various animal studies that model pathologies of, e.g., the gastrointestinal tract [14,15,16], liver [17,18], skeletal muscle [19,20,21], and peripheral neurons [22], both DMG and its sodium salt were shown to reduce oxidative stress damage by exhibiting scavenger activities against excess free radicals and to improve the utilization of oxygen [14,15,16,17,18,19,20,21,22,23]; hence, they can be conductive to the body’s redox status and its regeneration and repair. Moreover, DMG, acting as a methyl donor, has been suggested to improve immunity and enhance immune responses [24,25,26].

Despite these beneficial effects for maintaining the homeostasis of the body, to our knowledge, so far there is no report in the literature on the putative effects of DMG or its salt on the human skin, the largest organ of the body with constant cellular and tissue regeneration–rejuvenation–remodeling [27,28,29], and continuous exposition to stressor challenges from the environment [30,31,32,33]. Therefore, in this current proof-of-concept study, we aimed at assessing the effects of N,N-dimethylglycine sodium salt on different biological functions (proliferation, migration, growth factor synthesis, inflammatory processes) of cultured human epidermal keratinocytes. Among the available, quite versatile human skin models, e.g., in vitro epidermal cell cultures, reconstructed skin equivalents, and ex vivo full-thickness skin organ cultures [34,35,36,37], among which all have numerous advantages but also shortcomings, we carried out the current study on the human HaCaT keratinocyte cell line, as it is one of the most widely used cell lines in the field of experimental dermatology. Indeed, human epidermal HaCaT keratinocytes, cultured either in vitro or in reconstructed skin equivalents, have been extensively employed to model, e.g., keratinocyte proliferation and migration [38,39,40], the complex processes of re-epithelization in skin wound healing [41,42], growth factor synthesis and release [43,44], epidermal senescence and aging [45,46], as well as numerous inflammatory conditions [47,48,49].

Here, we report for the first time that DMG-Na promotes proliferation, migration, and the release of specific growth factors from epidermal keratinocytes. Moreover, we provide the first evidence that DMG-Na exerts profound anti-inflammatory and antioxidant effects, as assessed in three different human keratinocyte models, mimicking microbial and allergic contact dermatitis [50,51,52], inflammatory skin diseases such as psoriasis [36,53], and UVB irradiation-induced solar dermatitis [54,55,56].

Therefore, DMG-Na may constitute a novel, promising active compound in various skin-targeting topical formulations for promoting epidermal homeostasis, protection, regeneration, and repair. Furthermore, our findings provide new insights on the potential therapeutic applications of DMG-Na in the management of certain inflammatory conditions of the skin, and possibly even its appendages.

## 2. Results

### 2.1. DMG-Na Promotes Viability, Cell Proliferation, Migration, and the Release of Specific Growth Factors in Human Epidermal Keratinocytes

#### 2.1.1. DMG-Na Maintains Cellular Viability of Human Epidermal HaCaT Keratinocytes

First, the effects of three different concentrations of DMG-Na (0.00005%, 0.0005%, and 0.005%) on the cellular viability of human HaCaT keratinocytes were assessed by the colorimetric MTT assay in multiple independent experiments. Notably, even at the highest concentration (0.005%), DMG-Na did not compromise the cellular viability of the cells up to 72 h of treatment (Figure 1A). Instead, DMG-Na appeared to (at least tendentially) increase the viable cell number, especially at the 72 h time point. These findings suggested that DMG-Na may stimulate the proliferation of human HaCaT keratinocytes.

#### 2.1.2. DMG-Na Increases the Number of Ki67-Positive Cells and Upregulates the mRNA-Level Expression of Ki67

In order to investigate whether DMG-Na has pro-proliferative effects, fluorescence immunolabeling and qRT-PCR studies assessing the expression of the nuclear marker and indicator of cellular proliferation, Ki67, were performed [57]. Notably, DMG-Na increased (albeit only tendentially, due to the large standard error) the number of Ki67-positive cells after both 24 and 48 h of treatment (Figure 1B,C and Appendix A). Moreover, 0.0005% and 0.005% DMG-Na significantly (*p* < 0.05, ANOVA) upregulated the mRNA-level expression of Ki67 after 24 h of administration (Figure 1D). These data collectively further argued for the proliferation-stimulating action of DMG-Na.

#### 2.1.3. DMG-Na Promotes Migration of Human Epidermal HaCaT Keratinocytes

To assess whether the proliferation-promoting effects of DMG-Na are accompanied by a stimulation of cellular migration, further experiments were performed using the scratch wound assay [42]. For these experiments, the fetal bovine serum (FBS) concentration of the culturing media was reduced to 5% to decrease the rate of cellular proliferation (albeit without compromising cell viability and attachment). In addition to its effects on cell proliferation, DMG-Na also promoted human epidermal keratinocyte migration (Figure 1E,F and Appendix A). As revealed by the area under the curve (AUC) statistical analysis of the pooled wound-closure curves (Figure 1F), 0.005% DMG-Na significantly accelerated the scratch wound closure. This result reflects an augmented migration (AUC_Control_ = 2402 ± 56.8, AUC_DMG 0.005%_ = 1925 ± 119, mean ± SD, *p* < 0.05, ANOVA), which is also apparent from the photomicrographs taken during the individual experiments (Figure 1E and Appendix A).

#### 2.1.4. DMG-Na Upregulates the Synthesis and Release of Specific Growth Factors in Human Epidermal HaCaT Keratinocytes

We next assessed the effects of DMG-Na on the synthesis and release of selected growth factors. As shown in Figure 2, DMG-Na treatment of human epidermal keratinocytes significantly (*p* < 0.05, ANOVA) upregulated the synthesis (0.00005%, 0.0005%, and 0.005%), while only the 0.005% DMG-Na concentration upregulated the release of vascular endothelial growth factor (VEGF) (Figure 2A,B). Further, 0.005% DMG-Na tendentially (*p* = 0.074 for qRT-PCR, *p* = 0.071 for ELISA) stimulated the synthesis and release of granulocyte-macrophage colony-stimulating factor (GM-CSF) (shown in Appendix A). When assessing the expressions of other growth factors, we found that DMG-Na did not upregulate the mRNA-level synthesis of genes encoding transforming growth factor (TGF)β1 and TGFβ2 (Appendix A). Further, expression levels of genes encoding hepatic growth factor (HGF) and epidermal growth factor (EGF) were below the detection limit, both in the control and the DMG-Na-treated keratinocytes, also indicating a lack of upregulation by DMG-Na.

### 2.2. DMG-Na Exerts Anti-Inflammatory Effects in Three In Vitro Keratinocyte Models of Various Inflammatory Skin Conditions

#### 2.2.1. DMG-Na Exerts Anti-Inflammatory Effects in an In Vitro Keratinocyte Model of Microbial and Allergic Contact Dermatitis

In order to assess whether the above growth-promoting effects of DMG-Na, arguing for the stimulation of epidermal regeneration and repair, are accompanied by protective effects against inflammatory skin cell conditions, we also investigated the impact of DMG-Na in three different in vitro inflammation models of cultured human keratinocytes. Since the two highest tested DMG-Na concentrations (0.005% and 0.0005%) were most effective in the above-presented experiments, we focused on these two concentrations for the following assessments.

First, the potential anti-inflammatory effects of DMG-Na were assessed on toll-like receptor (TLR)-3 activation-induced cultured human HaCaT keratinocytes, a well-established in vitro model of microbial and allergic contact dermatitis [50,51,52]. As assessed by qRT-PCR (Figure 3A), both tested concentrations of DMG-Na prevented the action of the inflammation-inducer TLR3-agonist polyinosinic:polycytidylic acid (Poly-IC, 20 μg/mL) to upregulate the expression of various inflammatory markers. Notably, the anti-inflammatory effects of DMG-Na were statistically significant (*p* < 0.05, ANOVA) in the cases of genes encoding interleukin (IL)1α (IL1A), IL1β (IL1B), IL8 (IL8), and TNFα (TNFA), while the effect on IL6 (IL6) was not significant.

As measured by ELISA (Figure 3B), DMG-Na in both concentrations significantly (*p* < 0.05, ANOVA) inhibited the action of Poly-IC to induce the inflammation-associated release of IL1β, while only the higher concentration (0.005%) of DMG-Na had a statistically significant effect on the release of IL6 proteins. However, no effect of DMG-Na was observed on the Poly-IC-induced release of IL8, despite the robust and significant effect on the IL8 gene expression, as measured by qRT-PCR (Figure 3A).

#### 2.2.2. DMG-Na Exerts Anti-Inflammatory Effects in an In Vitro Keratinocyte Model of Various Inflammatory Skin Diseases Such as Psoriasis

Next, the putative anti-inflammatory effects of DMG-Na were assessed on cultured human HaCaT keratinocytes stimulated with the combination of 25 ng/mL of interferon-γ (INFγ) + 25 ng/mL of TNFα, a well-established in vitro model of inflammatory skin diseases including, but not limited to, psoriasis [36,53].

As shown in Figure 4A,B, DMG-Na prevented the effect of the INFγ + TNFα cocktail to upregulate the mRNA-level expression (qRT-PCR) and protein-level secretion (ELISA) of the inflammatory cytokines IL6, IL8, and C-X-C motif chemokine ligand 10 (CXCL10), also known as INFγ-induced protein 10. Notably, except for the protein-level secretion of IL6, these effects were statistically significant for 0.005% DMG-Na (*p* < 0.05, ANOVA), and in the case of IL8 (both mRNA and protein levels) for 0.0005% DMG-Na as well.

#### 2.2.3. DMG-Na Exerts Anti-Inflammatory and Antioxidant Effects in an In Vitro Keratinocyte Model of UVB Irradiation-Induced Solar Dermatitis

Finally, the potential anti-inflammatory effects of DMG-Na were assessed on UVB-irradiated, cultured human HaCaT keratinocytes to mimic solar dermatitis in vitro [54,55,56]. In this model, DMG-Na significantly (*p* < 0.05, ANOVA) prevented the effect of UVB to upregulate the expressions of IL6 (0.005% DMG-Na) and TNFA (both concentrations) genes; however, only tendential prevention was observed for IL1A and IL1B (both concentrations), as well as IL8 (0.005% DMG-Na) (Figure 5A). Notably, both concentrations of DMG-Na significantly (*p* < 0.05, ANOVA) prevented the action of UVB to induce the secretion of IL6 and IL8 proteins (Figure 5B). In this model, we also assessed whether DMG-Na can also mitigate the well-established [58,59,60] pro-oxidation effect of UVB irradiation. Both investigated concentrations significantly (*p* < 0.05, ANOVA) prevented the action of UVB irradiation to induce the production of reactive oxygen species (ROS) (Figure 5C).

## 3. Discussion

The findings of our in vitro investigation collectively demonstrated that DMG-Na exerted a positive impact on crucial parameters for epidermal biology by maintaining cell viability, promoting cellular proliferation and migration, and inducing both the synthesis and release of certain growth factors (VEGF, GM-CSF) in cultured immortalized human epidermal keratinocytes. Further, we provided the first evidence that DMG-Na exerts profound anti-inflammatory actions in all three in vitro models of epidermal inflammatory conditions employed, which are accompanied by its antioxidant effects against UVB irradiation.

The presented data might have numerous impactful cutaneous perspectives. It was previously shown that DMG, as part of the endogenous metabolic homocysteine pathway in plants and animals [1,2], and its sodium salt markedly modulate numerous mitochondria functions and energy output [18,19,20,21]. Indeed, it was found that DMG-Na activates the mitochondrial metabolic network of nuclear factor erythroid 2-related factor 2/sirtuin 1/peroxisome proliferator-activated receptor-γ coactivator-1α (NRF2/Sirtuin 1/PGC1α) in other tissues (skeletal muscle, liver), which established the most robust basis for its beneficial effects. Importantly, these pathways are also involved in preventing premature and/or pathological aging of the skin [61,62,63]. Likewise, VEGF (released from epidermal keratinocytes upon DMG-Na treatment) was recently implicated as a key growth factor promoting skin rejuvenation [64]. Therefore, it is proposed (and hence to be assessed in further studies) that DMG-Na may not only boost overall epidermal health and fitness (based on its “anabolic” effects presented in this study) but most probably also acts as a novel rejuvenating and anti-aging active compound by subsequently releasing VEGF.

Another skin condition where great potential for DMG-Na could be realized is the management of wound healing. Indeed, we found that DMG-Na stimulates proliferation and migration of epidermal keratinocytes, key events of re-epithelization after wounding [65,66]. Further, we also showed that DMG-Na upregulates the synthesis and release of VEGF and GM-CSF, positive master regulators of skin wound healing [67,68,69,70,71]. Since DMG was previously shown to promote proliferation and overall activation of human lymphocytes [24,25], another key event of wound healing, the beneficial effects of DMG should be explored in future skin wound treatments.

Based on the above, we also propose that DMG-Na could be beneficial in maintaining not only epidermal, but also hair (follicle) health. Hair follicles are considered as the “brain of the skin” as they control, via their cell reservoirs and by releasing a plethora of soluble mediators, practically all cutaneous functions [72,73,74]. With respect to our current data, it is intriguing to note that VEGF (that was found to be released by DMG-Na in our in vitro epidermal model) was previously shown to markedly promote hair growth [75,76,77] and to be involved in mediating the action of minoxidil, one of the few approved anti-alopecia drugs [78], thereby establishing a concept for DMG-Na as a putative new active against hair loss.

Possibly the most striking finding of our work was the identification of the robust, in most cases both biologically and statistically significant, anti-inflammatory effects of DMG-Na, phenomena that had never been previously investigated in any tissue. Of further importance, these actions of DMG-Na were observed in all three in vitro keratinocyte systems, modeling a wide array of inflammatory skin diseases, irrespective of whether the inflammation was induced by TLR3 stimulation, the combination of INFγ and TNFα, or UVB irradiation. Since, in these models, human epidermal keratinocytes engage distinct and/or only partially overlapping intracellular signaling pathways to upregulate expression and secretion of a myriad of pro-inflammatory cytokines [36,55,79], and since DMG-Na exhibited a remarkable effect in mitigating the upregulation of most of the investigated inflammatory markers in all three models, it can be concluded that DMG-Na exerts “universal” (i.e., model-independent) anti-inflammatory actions.

Further, by using the classical UVB irradiation-induced model system, we also showed that DMG-Na significantly counteracted the effect of UVB irradiation to upregulate ROS production of epidermal keratinocytes. This corroborates well with previous studies that reported the profound antioxidant and scavenger actions of DMG and its sodium salt in practically all (extra-cutaneous) tissues measured so far [14,15,16,17,18,19,20,21,22,23], including the gastrointestinal tract [14,15,16], liver [17,18], skeletal muscle [19,20,21], and peripheral neurons [22].

Further studies are now invited to describe the molecular mechanism of action of DMG-Na. Along this line, in our preliminary experiments, we found that DMG-Na induced a dose-dependent increase of the intracellular Ca^2+^ concentration of human epidermal keratinocytes (microfluorimetric Ca-imaging data obtained in two independent experiments are shown in Appendix A). Elevation of the intracellular Ca^2+^ concentration is one of the key intracellular signaling pathways that regulate practically all keratinocytes’ functions, including, but not limited to, proliferation, differentiation, migration, wound healing, growth factor release, inflammatory and immune signaling pathways, etc. [80,81,82,83,84,85,86]. Therefore—albeit further experiments are needed to uncover how the intracellular Ca^2+^ concentration elevated by DMG-Na is mechanistically linked to such signaling systems as, e.g., certain inflammatory signal transduction pathways (NF-κB, MAPK, or JAK-STAT) or the release of VEGF and other mediators—it is highly possible that the quite versatile effects (stimulation of proliferation, migration, and growth factor release; anti-inflammatory and antioxidant action) of DMG-Na are mediated (or at least closely linked) to its action on elevating the intracellular level of the ‘universal’ signal transducer Ca^2+^.

Taken together, our preclinical findings presented here collectively argue for the versatile effects of DMG-Na on promoting epidermal proliferation, regeneration, protection, as well as epithelial and inflammatory repair. Further clinical studies are currently under investigation to assess whether topically applied DMG-Na can be therapeutically employed in the future management of skin and hair conditions characterized by impaired cellular growth, repair, and dysregulated oxidative metabolic status (e.g., skin aging, atrophy, acute or chronic wounds, hair loss of any kind, among others), as well as various inflammatory skin diseases (e.g., psoriasis, allergic contact dermatitis, solar irritation, and dermatitis).

## 4. Materials and Methods

### 4.1. Materials

The sodium salt of N,N-dimethylglycine was obtained from Eastman Chemical Company (Eastman Enhanz^TM^, Kingsport, TN, USA). Human TNFα and IFNγ recombinant proteins were purchased from Thermo Fisher Scientific (Waltham, MA, USA; Cat. No. PHC3011 and PHC4031), whereas Poly-IC was bought from Merck KGaA (Darmstadt, Germany; Cat. No. P0913). All agents were dissolved in nuclease-free water (Thermo Fisher Scientific; Cat. No. AM9930).

### 4.2. Cell Culturing

Spontaneously immortalized human HaCaT epidermal keratinocytes were cultured in Dulbecco’s Modified Eagle Medium supplemented with 10% (*v*/*v*) FBS (or 5% (*v*/*v*) for the migration assay), 1% (*v*/*v*) Penicillin–Streptomycin mixture, and 0.5% (*v*/*v*) Amphotericin B (all from Thermo Fisher Scientific; Cat. No. 31966021, 10500064, 15140122, and 15290026, respectively), at 37 °C in a humidified, 5% CO_2_-containing atmosphere. The Ca^2+^ concentration of the cell culturing medium was 264 mg/mL (1.8 mM). As shown in our previous publication [87], this medium allows the constant proliferation of pre-confluent HaCaT keratinocytes. The current study was performed on pre-confluent (hence proliferating) HaCaT keratinocytes. The medium was substituted every other day and cells were sub-cultured at 70–80% confluence [88,89].

### 4.3. MTT Assay

The MTT assay was used to measure cellular metabolic activity as an indicator of cell viability, proliferation, and cytotoxicity. For the experiments, HaCaT keratinocytes (5 × 10^3^ cells/well) were cultured in 96-well plates and treated as indicated with DMG-Na for 24, 48, and 72 h; then, cells were incubated with 0.5 mg/mL of MTT (Merck KGaA; Cat. No. M5655) for 2 h. The insoluble formazan crystals were dissolved using a solubilizing solution containing 81% (*v*/*v*) 2-propanol, 9% (*v*/*v*) 1M HCl, and 10% (*v*/*v*) Triton-X100 (all from Merck KGaA; Cat. No. I9516, 320331 and X100), and the resulting colored solution was quantified by measuring absorbance at 540 nm using a Halo LED 96-microplate reader (Dynamica Scientific Ltd., Livingston, UK) [90,91,92].

### 4.4. Immunocytochemistry

HaCaT keratinocytes (3 × 10^4^ cells/10 mm circular glass coverslips) treated with DMG-Na were fixed with freshly prepared 1% paraformaldehyde (Merck KGaA; Cat. No. P6148) for 10 min at room temperature and then washed with phosphate-buffered saline (PBS, Merck KGaA; Cat. No. P3813), followed by a post-fixation step with −20 °C ethanol:acetic acid (2:1) for 5 min. In the permeabilization step, cells were incubated with 0.25% Triton X-100 in PBS for 10 min and were then blocked with Antibody Diluent (Thermo Fisher Scientific; Cat. No. 003118) for 30 min. Cells were incubated with mouse anti-human Ki67 antigen primary antibody at 1:100 dilution (Dako, Agilent, Santa Clara, CA, USA; Cat. No. M7240) at 4 °C overnight. Following the appropriate washing step with PBS, cells were labeled with the secondary antibody, Alexa Fluor 568 goat anti-mouse IgG (Thermo Fischer Scientific; Cat. No. A-11004), at 1:500 dilution for 45 min at room temperature. Superfluous secondary antibodies were washed away with PBS and nuclei were counterstained with 4′,6-diamidino-2-phenylindole (DAPI, Merck KGaA; Cat. No. D9542). After another washing step, cells on slides were mounted with Fluoromount-G^®^ (SouthernBiotech, Birmingham, AL, USA; Cat. No. 0100-01) and images were taken by a Nikon A1 confocal microscope (Nikon, Tokyo, Japan). During image analysis, cells positive for Ki67 were counted in multiple visual fields and were normalized to the number of nuclei, i.e., the number of DAPI+ cells, by using the Image J Fiji software version 2.9.0. (National Institutes of Health, Bethesda, MD, USA) [89,93].

### 4.5. Scratch Wound-Closure Migration Assay

HaCaT keratinocytes were seeded at a density of 2 × 10^4^ cells/chamber in 2-Well Culture Inserts (Ibidi GmbH, Gräfelfing, Germany; Cat. No. 80209). After reaching 100% confluence, a cell-free gap of 500 μm was created by removal of the plastic inserts, and the cells were treated with DMG-Na, as indicated in the above culturing medium containing 5% FBS. Images were then taken of the cultures at different time points (0, 16, 20, 24, and 40 h) using an Olympus IX81 inverted microscope (Olympus Corporation, Tokyo, Japan), and the size of the scratch wounds (the decrease of which corresponds to the degree of cell migration) were measured by Image J Fiji software version 2.9.0. For the analysis, wound-closure curves were generated and area under the curve (AUC) values were calculated, where the AUC value is inversely proportional to the velocity of wound closure [42,94].

### 4.6. Inflammatory Model Systems

#### 4.6.1. In Vitro Keratinocyte Model of Microbial and Allergic Contact Dermatitis

TLR-3 activation on in vitro-cultured human HaCaT keratinocytes results in such pro-inflammatory alterations which well-model the processes of microbial and allergic contact dermatitis [50,51,52]. For this, keratinocytes were seeded at a density of 3 × 10^5^ cells/well in 6-well plates. The next day, the medium was renewed, and keratinocytes were pre-treated with DMG-Na for 1 h, as indicated, and then stimulated with a TLR3 ligand, 20 μg/mL of Poly-IC, for an additional 23 h. After incubation, supernatants and cells were harvested and processed for ELISA and qRT-PCR determinations, respectively, of various inflammation markers.

#### 4.6.2. In Vitro Keratinocyte Model of Inflammatory Skin Diseases Such as Psoriasis

Treatment of cultured human HaCaT keratinocytes with a defined combination of IFNγ and TNFα is a well-established in vitro model for studying cellular changes of various inflammatory skin diseases, e.g., psoriasis, etc. [36,53]. For this, HaCaT cells were seeded at a density of 3 × 10^5^ cells/well into 6-well plates. After cell adherence, the medium was replaced with a fresh conventional culture medium with or without DMG-Na. After 1 h, cells were stimulated with a combination of 25 ng/mL of IFNγ and 25 ng/mL of TNFα, as previously described [36,95], and 23 h later, supernatants were collected and cells were harvested for qRT-PCR and ELISA analyses of the expression and release of various pro-inflammatory cytokines.

#### 4.6.3. UVB Irradiation-Induced Inflammation and Oxidative Stress Model

Keratinocytes (3 × 10^5^ cells/well) were seeded in 6-well plates and treated with DMG-Na for 1 h, as indicated. After this pre-treatment and before irradiation, the medium was replaced with 800 μL/well of Sebomed Basal Medium (Merck KGaA; Cat. No. F8205). Lids were removed and cells were irradiated with a total dose of 40 mJ/cm^2^ of UVB (312 nm) by using a narrow-band, microprocessor-controlled UV irradiation instrument (Bio-Sun, Vilber Lourmat, Marne-la-Vallée, France). Control samples were sham-irradiated under the same conditions. Immediately after UVB irradiation, the medium was replaced with the conventional culture medium of the cells with or without DMG-Na. Following a 6 h incubation period, cells and supernatants were collected for qRT-PCR and ELISA to determine the cytokine expression and release.

As UVB irradiation of epidermal keratinocytes is not only a pro-inflammatory but also an oxidative stress [54,55,56,58,59,60], in this model, we also assessed the putative effects of DMG-Na on the UVB-induced ROS production. For this, we employed the DCFDA/H2DCFDA-Cellular ROS Assay Kit (Abcam, Cambridge, UK; Cat. No. ab113851) according to the manufacturer’s instructions. In brief, cells were seeded at a density of 2.5 × 10^4^ cells/well in a black, clear-bottom, 96-well plate, and treated as indicated. Immediately after irradiation with 100 mJ/cm^2^ of UVB (Bio-Sun, Vilber Lourmat), the medium was discharged, and cells were washed with 1X Buffer and then incubated with 20 μM of 2′,7′-dichlorofluorescein diacetate (DCFDA) solution for 45 min at 37 °C, protected from light. After the incubation, the DCFDA solution was removed, 1X Buffer was added to the cells, and fluorescence was measured at an excitation/emission of 485/535 nm using a FlexStation II (Molecular Devices, Sunnyvale, CA, USA) fluorescence microplate reader.

### 4.7. RNA Isolation and Reverse Transcription

Total RNA was isolated using the TRI Reagent^®^ (Molecular Research Center, Inc., Cincinnati, OH, USA; Cat. No. TR 118) according to the manufacturer’s protocol, and quality-checked with a NanoDrop 2000 spectrophotometer (Thermo Fisher Scientific). Next, DNase I treatment was performed, and 1000 ng of total RNA was reverse-transcribed into complementary DNA (cDNA) by using the High-Capacity cDNA Reverse Transcription Kit (both obtained from Thermo Fisher Scientific; Cat. No. AM2222 and 4368813) on an Applied Biosystems 2720 Thermal Cycler (Thermo Fisher Scientific), according to the manufacturer’s protocols [88,89,90,91,92,96].

### 4.8. Quantitative Real-Time Polymerase Chain Reaction (qRT-PCR)

Quantitative real-time PCR was performed on a Light Cycler^®^ 480 Instrument II (Roche Life Science, Basel, Switzerland) by using the TaqMan 5′ nuclease assays and the TaqMan Gene Expression Master Mix protocol (both from Thermo Fisher Scientific; Cat. No. 4331182 and 4369016). The following assays were used: CXCL10 (Hs99999049_m1), GM-CSF (Hs00929873_m1), IL-1α (Hs00174092_m1), IL-1β (Hs00174097_m1), IL-6 (Hs00985639_m1), IL-8 (Hs00174103_m1), Ki67 (Hs01032443_m1), TNFα (Hs00174128_m1), VEGF (Hs00900055_m1), TGFβ1 (Hs00171257_m1), TGFβ2 (Hs00234244_m1), EGF (Hs01099999_m1), and HGF (Hs00300159_m1). As endogenous controls, transcripts of actin-β (ACTB, Hs99999903_m1), glyceraldehyde 3-phosphate dehydrogenase (GAPDH, Hs99999905_m1), or peptidylprolyl isomerase A (PPIA, Hs99999904_m1) were determined, respectively. The amount of the abovementioned transcripts was normalized first to the expression of the relevant housekeeping gene, then to the expression of the vehicle control using the 2^−ΔΔCT^ method [92,97]. qRT-PCR analysis was performed on samples treated for 24 h.

### 4.9. Enzyme-Linked Immunosorbent Assay (ELISA)

Keratinocytes were treated as indicated for 6 or 24 h (for determining pro-inflammatory markers) or 48 h (for determining growth factors), then supernatants were collected and the released amounts of interleukin (IL)-1β, IL-6, or IL-8 (BD OptEIA ELISA Kits, Becton Dickinson, BD Biosciences, Franklin Lakes, NJ, USA; Cat. No. 557953, 555220 and 555244) and/or CXCL10, VEGF, or GM-CSF (Merck KGaA; Cat. No. RAB0119, RAB0507 and RAB0100) were determined according to the manufacturer’s protocols. Absorbance was measured at 450 nm using a Halo LED 96-Microplate reader (Dynamica Scientific Ltd.). The amount of cytokines in the supernatant was expressed in pg/mL, calculated from standard curves [88,89,91,92].

### 4.10. Ca-Imaging—Microfluorimetric Measurements of Intracellular Ca Concentration ([Ca^2+^]_i_)

HaCaT keratinocytes were seeded in 96-well, black-well/clear-bottom plates (Greiner Bio-One, Kremsmünster, Austria; Cat. No. 655090) at a density of 2 × 10^4^ cells/well in keratinocyte medium supplemented as above and cultured at 37 °C for 24 h. The cells were then loaded with the cytoplasmic calcium indicator 2 μM Fluo-4 AM (Thermo Fisher Scientific; Cat. No. F14201) at 37 °C for 1 h in Hank’s balanced salt solution (136.89 mM NaCl, 5.55 mM glucose, 5.36 mM KCl, 4.17 mM NaHCO_3_, 1.26 mM CaCl_2_, 0.49 mM MgCl_2_ × 6 H_2_O, 0.34 mM Na_2_HPO_4_ × 2 H_2_O, 0.44 mM KH_2_PO_4_, 0.24 mM MgSO_4_ × 7 H_2_O, pH 7.2; all from Merck KGaA) containing 2.5 mM of Probenecid and 1% bovine serum albumin (both from Merck KGaA). The cells were washed and the plates were then placed into a FlexStation II384 Fluorescence Imaging Plate Reader (Molecular Devices, Sunnyvale, CA, USA), and changes in [Ca^2+^]_i_ (reflected by changes in fluorescence, λ_EX_ = 494 nm, λ_EM_ = 516 nm) induced by various concentrations of DMG-Na were recorded in each well (during the measurement, cells in a given well were exposed to only one given concentration of the agent). Experiments were performed in quadruplicates and the averaged data were used in the calculations [98,99].

### 4.11. Statistical Analysis

For each read-out, *n* = 3–4 independent experiments were performed, with multiple intra-experimental technical repeats. Data obtained in the independent experiments were pooled. For normally distributed data, groups were compared using ordinary one-way ANOVA, followed by Dunnett’s post hoc test. When data distribution was not normal, the Kruskal–Wallis nonparametric test with Dunn’s multiple comparison test were performed using GraphPad Prism version 8.0.1. for Windows (GraphPad Software, San Diego, CA, USA). *p*-values < 0.05 were regarded as significant differences. Graphs were plotted by using OriginPro 8.6 software (OriginLab Corporation, Northampton, MA, USA). The results are shown as mean ± standard errors of the mean (SEM) or mean ± standard deviation (SD).

## 5. Conclusions

Our study identified DMG-Na as a highly promising novel active ingredient that—by stimulating epidermal proliferation and exerting robust anti-inflammatory as well as antioxidant actions—maintains overall epidermal fitness, promotes regeneration and repair, and applies protective functions. Therefore, further preclinical and clinical studies are ongoing to systematically assess the beneficial effects of DMG-Na in the management of relevant skin conditions.

## Figures and Tables

**Figure 1 ijms-24-11264-f001:**
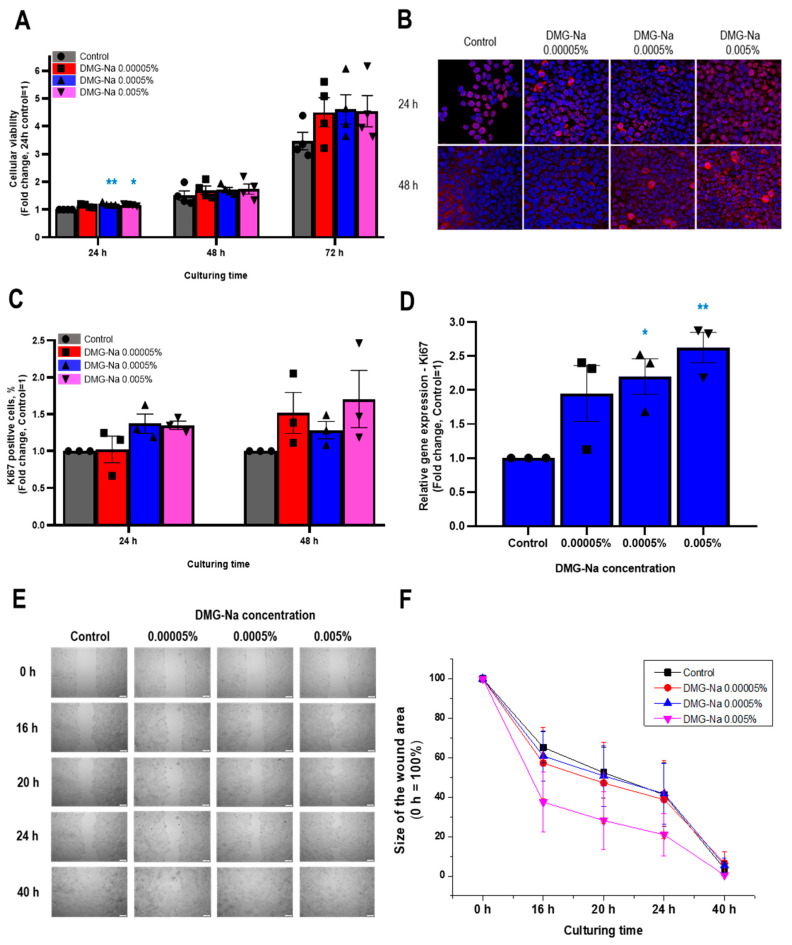
Effects of DMG-Na on the cellular viability, proliferation, and migration of human epidermal HaCaT keratinocytes. (**A**) Effects of various concentrations of DMG-Na on the viability of HaCaT keratinocytes were assessed by the MTT assay after 24, 48, and 72 h of treatment. Pooled data of independent experiments (*n* = 4) are shown. (**B**–**D**) Effects of various concentrations of DMG-Na on the expression of the proliferation marker Ki67 after 24 and 48 h of treatment. Representative images of Ki67 immunofluorescence labeling (Ki67: purple, DAPI: blue) of Experiment 1 (**B**) (the other 2 experiments are presented in Appendix A). Pooled data of these independent experiments (*n* = 3) on determining the percentage of Ki67 immuno-positive cells (**C**). Pooled data of independent experiments (*n* = 3) on determining the expression of mRNA transcripts of Ki67 by qRT-PCR (**D**). (**E**,**F**) Effects of DMG-Na on scratch wound closure (i.e., migration) of human epidermal HaCaT keratinocytes, as assessed at different time points. Representative photomicrographs of Experiment 1 (**E**) (the other 2 experiments are presented in Appendix A). Pooled data of wound-closure curves of independent experiments (*n* = 3) (**F**). Data are presented as mean ± SEM. * *p* < 0.05, ** *p* < 0.01, compared to the control.

**Figure 2 ijms-24-11264-f002:**
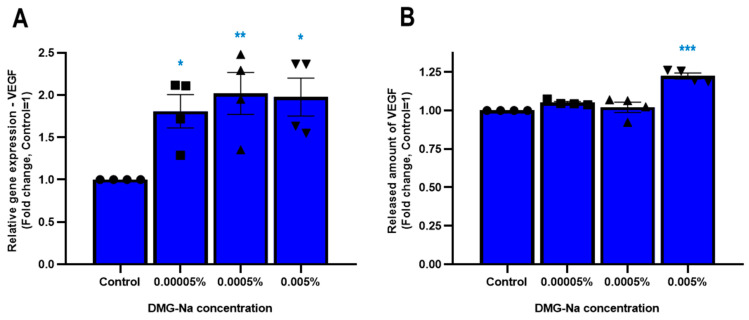
Effects of DMG-Na on the synthesis and release of vascular endothelial growth factor in human epidermal HaCaT keratinocytes. Effects of various concentrations (0.00005%, 0.0005%, and 0.005%) of DMG-Na on the mRNA-level ((**A**) qRT-PCR assay) and protein-level release ((**B**) ELISA) of vascular endothelial growth factor (VEGF) of HaCaT keratinocytes were assessed after 24 h of qRT-PCR and 48 h of ELISA treatment. Each panel shows pooled data of independent experiments (*n* = 4). Mean ± SEM. Data points are shown as ● for Control, ■ for 0.00005% DMG-Na, ▲ for 0.0005% DMG-Na and ▼ for 0.005% DMG-Na. * *p* < 0.05, ** *p* < 0.01, *** *p* < 0.001 by ANOVA, compared to the control.

**Figure 3 ijms-24-11264-f003:**
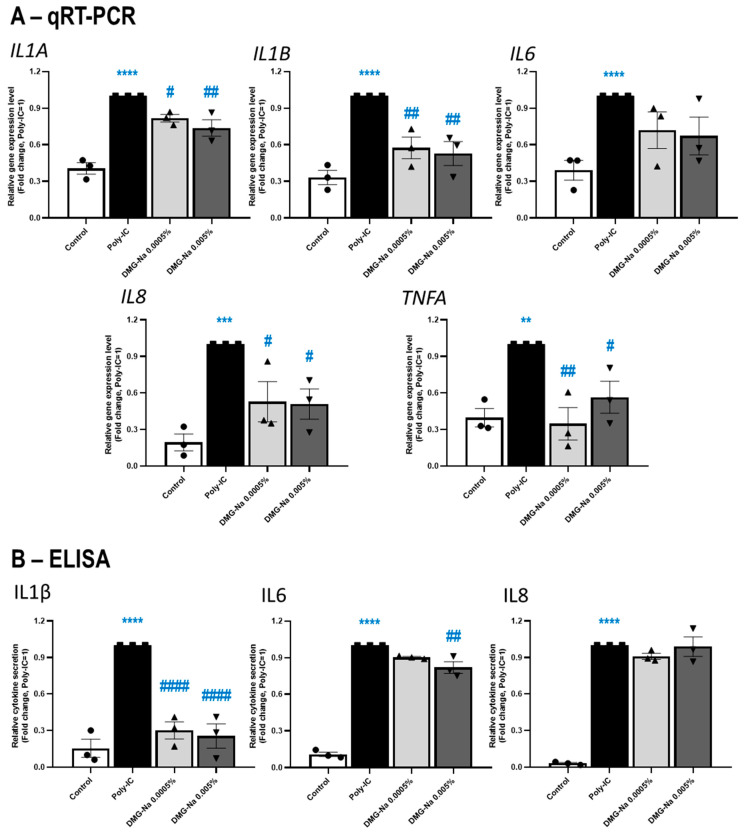
Effects of DMG-Na on the synthesis and release of inflammation markers in a keratinocyte model of microbial and allergic contact dermatitis. Effects of two different concentrations of DMG-Na on the mRNA-level ((**A**) qRT-PCR assay) and protein-level release ((**B**) ELISA) of various inflammatory cytokines of HaCaT keratinocytes were assessed after 24 h. Cells were pre-treated with DMG-Na for 1 h and then inflammation was induced by the administration of the TLR3 activator Poly-IC (20 μg/mL). Each panel shows pooled data of independent experiments (*n* = 3). Data were normalized to the Poly-IC group and are presented as mean ± SEM fold change values (where Poly-IC = 1). Data points are shown as ● for Control, ■ for Poly-IC, ▲ for 0.0005% DMG-Na and ▼ for 0.005% DMG-Na. * Represents statistical differences when compared to the control, whereas # represents statistical differences when compared to the Poly-IC group. ^#^
*p* < 0.05, **,^##^
*p* < 0.01, *** *p* < 0.001, ****,^####^
*p* < 0.0001 by ANOVA.

**Figure 4 ijms-24-11264-f004:**
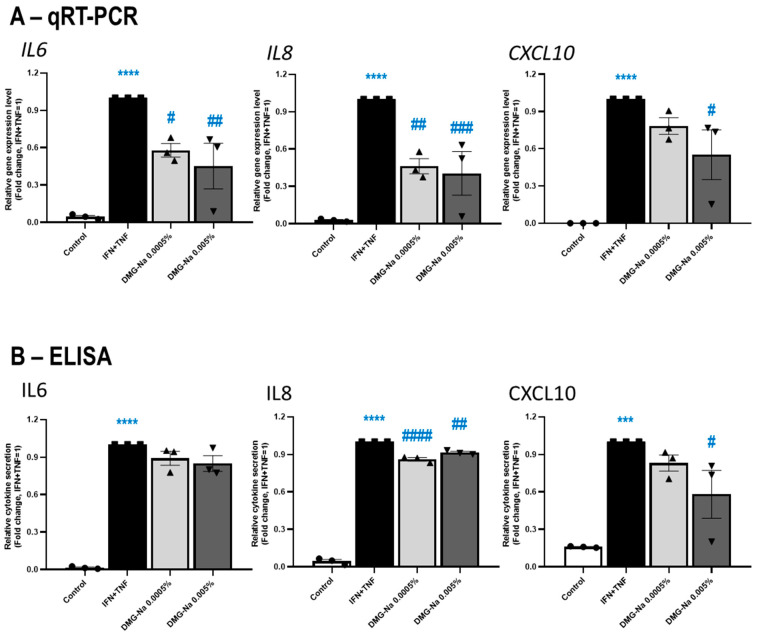
Effects of DMG-Na on the synthesis and release of inflammation markers in a keratinocyte model of psoriasis and atopic dermatitis. Effects of two different concentrations (0.0005% and 0.005%) of DMG-Na on the mRNA-level ((**A**) qRT-PCR assay) and protein-level release ((**B**) ELISA) of various inflammatory cytokines of HaCaT keratinocytes were assessed after 24 h. Cells were pre-treated with DMG-Na for 1 h and then inflammation was induced by the administration of 25 ng/mL of INFγ + 25 ng/mL of TNFα (IFN + TNF). Each panel shows pooled data of independent experiments (*n* = 3). Data were normalized to the IFN + TNF group and are presented as mean ± SEM fold change values (where IFN + TNF =1). Data points are shown as ● for Control, ■ for IFN + TNF, ▲ for 0.0005% DMG-Na and ▼ for 0.005% DMG-Na. * Represents statistical differences when compared to the control, whereas # represents statistical differences when compared to the IFN + TNF group. ^#^
*p* < 0.05, ^##^
*p* < 0.01, ***,^###^
*p* < 0.001, ****, ^####^
*p* < 0.0001 by ANOVA.

**Figure 5 ijms-24-11264-f005:**
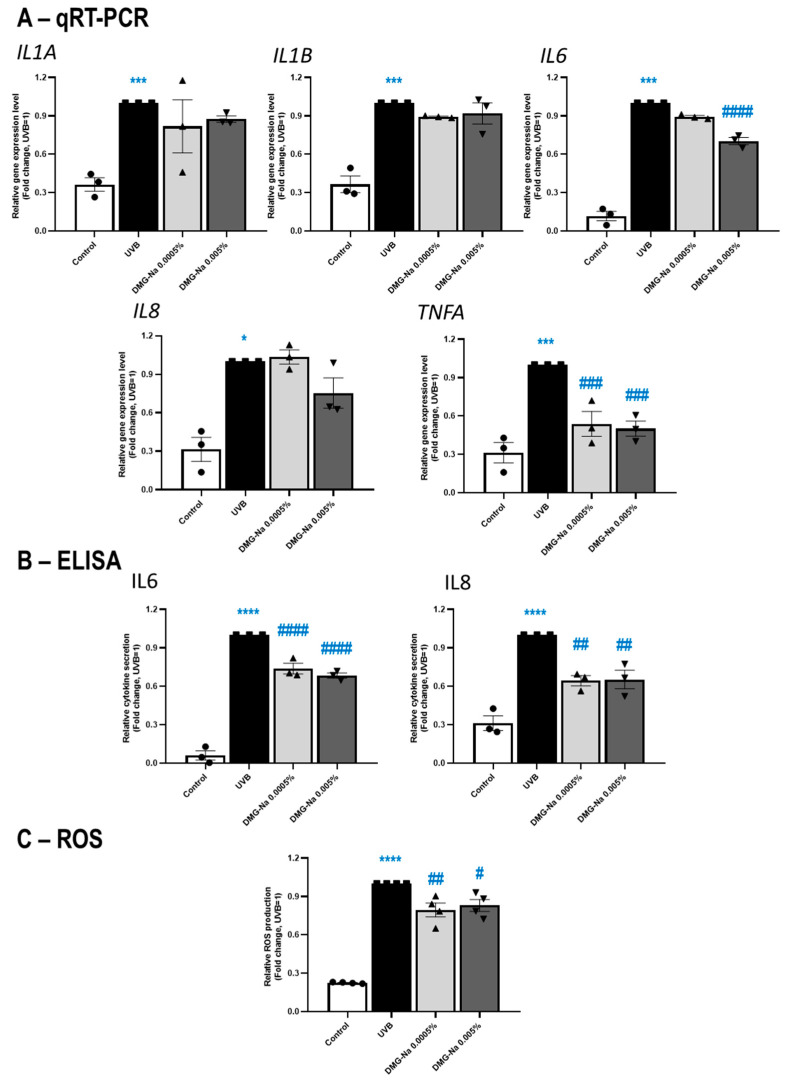
Effects of DMG-Na on the synthesis and release of inflammation markers as well as ROS in a keratinocyte model of UVB irradiation-induced solar dermatitis. Effects of two different concentrations (0.0005% and 0.005%) of DMG-Na on the mRNA-level ((**A**) qRT-PCR assay) and protein-level release ((**B**) ELISA) of various inflammatory cytokines of HaCaT keratinocytes were assessed after 6 h. Cells were pre-treated with DMG-Na for 1 h and then irradiated with a total dose of 40 mJ/cm^2^ of UVB (312 nm). Further, the effects of DMG-Na were also assessed on the UVB irradiation-induced ROS production ((**C**), 2′,7′-dichlorofluorescein diacetate-based assay). For this assay, cells were pre-treated with DMG-Na and vehicle for 1 h, irradiated with a total dose of 100 mJ/cm^2^ of UVB (312 nm), and the ROS production was immediately measured. Each panel shows pooled data of independent experiments (*n* = 3–4). Data were normalized to the UVB group and are presented as mean ± SEM fold change values (where UVB =1). Data points are shown as ● for Control, ■ for UVB, ▲ for 0.0005% DMG-Na and ▼ for 0.005% DMG-Na. * Represents statistical differences when compared to the control, whereas # represents statistical differences when compared to the UVB group. *,^#^
*p* < 0.05, ^##^
*p* < 0.01, ***,^###^
*p* < 0.001, ****,^####^
*p* < 0.0001 by ANOVA.

## Data Availability

The data presented in the study are available in the article and the Appendix A.

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
