# Peer review of "N,N-Dimethylglycine Sodium Salt Exerts Marked Anti-Inflammatory Effects in Various Dermatitis Models and Activates Human Epidermal Keratinocytes by Increasing Proliferation, Migration, and Growth Factor Release"

_ijms, 2023, doi:10.3390/ijms241411264_

Round 1

Reviewer 1 Report

Lendvai et al. present a study reporting the protective effect of N, N-Dimethylglycine Sodium Salt in HaCaT cells differently stimulated. Although the experimental design is quite straightforward and coherent, their paper has many gaps that must be filled.

Major issues:

It's not clear the reason why the Authors decided to use cultured keratinocytes as they did not discuss the pros and cons of their choice and an overview of available skin models is completely lacking. Moreover, as reported in Figure 1, SEM progressively increased during the time course experiment. Considering that one of the advantages offered by cell lines is the high reproducibility of the experimental conditions, I think that the Authors should give a satisfactory and acceptable explanation for this.

Another concern is the choice of IFN-gamma e TNF-alpha for mimicking psoriasis and atopic dermatitis. Whilst this setting can be an extrapolation for psoriasis, in the case of atopic dermatitis the Authors should stimulate cells with Th2 cytokines. Considering that in psoriasis and atopic dermatitis the involved cytokines are different, the Authors cannot assume that one model is suitable for mimicking both diseases. The cited ref 33 refers to the considered cytokines for psoriasis, vitiligo, and alopecia areata, not for atopic eczema.

Minor issues:

In the Introduction Section:

Authors should include more appropriate references for presenting the relationship between the skin and the environment

In the Result section:

The first sentence in paragraph 2.1.4 should be placed in the Introduction, while the qRT-PCR description should be moved to the Materials and Methods

In Figure 2, the reported variability is too elevated. I strongly suggest that the Authors remove this part.

In the Mat/Met Section

In paragraph 4.2 the Authors should specify the calcium concentration in the medium, strictly influencing cell differentiation.

In paragraph 4.6.1 the Authors wrote "stimulation of TLR3 activation". As it is redundant, please check the right expression.

In the Acknowledgments:

Please check

Author Response

Lendvai et al. present a study reporting the protective effect of N, N-Dimethylglycine Sodium Salt in HaCaT cells differently stimulated. Although the experimental design is quite straightforward and coherent, their paper has many gaps that must be filled.

Major issues:

1) It's not clear the reason why the Authors decided to use cultured keratinocytes as they did not discuss the pros and cons of their choice and an overview of available skin models is completely lacking. Moreover, as reported in Figure 1, SEM progressively increased during the time course experiment. Considering that one of the advantages offered by cell lines is the high reproducibility of the experimental conditions, I think that the Authors should give a satisfactory and acceptable explanation for this.

RESPONSE:

  1. A) About the skin model employed in our study

We appreciate for the very important comments of the Reviewer. As mentioned in the Introduction section, according to our best knowledge, this is the first study which aimed at assessing the biological effects of DMG-Na on the human skin; in other words, we could only hypothesize that DMG-Na, which was shown to modulate various biological functions of several organs, tissues and cell types, may also act on the skin.

Therefore, by means of ‘scientific practicality”, we decided to first employ the well-known, well-established and extensively characterized HaCaT human epidermal keratinocytes cell line. Indeed, the HaCaT keratinocyte cell line as it is one of the most widely used cell lines in the field of experimental dermatology – the popularity of HaCaT keratinocytes can perfectly be highlighted by the fact that the PubMed search with the term ’HaCaT’ resulted in >200 publications only in Int J Derm Sci. In the prior publications, HaCaT keratinocytes were shown to properly model such biological processes of human epidermal keratinocytes as model e.g. keratinocyte proliferation and migration, the complex processes of wound healing, growth factor synthesis and release, skin senescence and aging as well as numerous inflammatory skin conditions. By following the suggestion of the Reviewer, we included new parts to paragraph 2 of the Introduction to highlight the above.

We also fully agree with the Reviewer comments that the available pre-clinical skin models have definite pros and cons; yet, these are quite extensively reviewed in several prior publications. Therefore, we are not sure whether an overview (even a non-exhaustive one) of these models would be appropriate for our paper. Nevertheless, by following the suggestions of the Reviewer, under paragraph 2 of the Introduction section, we also mention the available skin models and introduce several new references which detail their advantages and shortcomings.

Finally, we would like to note that we have been assessing the effects of DMG-Na in additional human skin models, namely in in vitro cultures of other skin-derived cell types and also in different ex vivo skin organ cultures, and we hope to be able to repost on the results of these studies soon.

  1. B) About the large SEM values

Indeed, as also hinted by the Reviewer, the HaCaT human keratinocyte cell line allows for the characterization of human keratinocytes using a model that is reproducible and addresses issues such as short culture lifespan and variations that would otherwise be encountered by using primary keratinocytes, three dimensional organotypic skin cultures or skin organ cultures. Yet, according to our experience with a high number of different cell lines, independent experiments performed on different days (as was the case in our studies) always results in certain variations, even when using cell lines, as part of the well-known inherent complexity, hence heterogeneity, of any biological system. Another possible cause for the variations of the responses could be that the cellular effects of DMG-Na – which (as mentioned in the Introduction section) is a part of the endogenous homocysteine pathway and modulates various metabolic cellular processes (reference 1, 2, 14-21, 24-26) – strongly depend on the actual metabolic status of the cells which, even in cultures of cell lines, could be highly variable cell-by-cell.

Nevertheless, we believe that by i) performing sufficiently high number of independent experiments for each methods used; ii) employing several intra-experimental technical repeats for each individual experiment; and iii) the application of proper statistical methods, we managed to successfully circumvent the issues of inter-experimental variations, and that the presented data, both from the biological and statistical point-of-views, are sound and relevant.

  1. Another concern is the choice of IFN-gamma e TNF-alpha for mimicking psoriasis and atopic dermatitis. Whilst this setting can be an extrapolation for psoriasis, in the case of atopic dermatitis the Authors should stimulate cells with Th2 cytokines. Considering that in psoriasis and atopic dermatitis the involved cytokines are different, the Authors cannot assume that one model is suitable for mimicking both diseases. The cited ref 33 refers to the considered cytokines for psoriasis, vitiligo, and alopecia areata, not for atopic eczema.

RESPONSE:

Thank you very much for this important note. As mentioned in the paper, with the inflammation experiments, our aim was to reveal the potential anti-inflammatory effects of DMG-Na in various models in which the inflammation of keratinocytes was induced by very different stimuli; in other words, we were curious to find out whether the effects of DMG-Na depends or not in the actual model and the inducer stimuli. With the joint application of INFgamma + TNFalpha, key cytokines of a plethora of inflammatory skin conditions, our goal was to induce a ‘general inflammation’ of the cells where multiple (if not all) Th-coupled inflammatory responses are engaged and augmented. We fully agree with the Reviewer (and also with the other Reviewer and the Academic Editor) that atopic dermatitis is characterized by the key involvement of Th2 cytokines and pathways. Yet, numerous prior studies also highlight that additional (Th22, Th17 and even Th1) cytokines and immune signaling mechanism also play key, pathogenetic roles in the development of the disease, especially in certain subtypes (see e.g. in Brunner and Guttman-Yassky, J Allergy Clin Immunol. 2017 Apr;139(4S):S65-S76. doi: 10.1016/j.jaci.2017.01.011. and Tokura and Hayano Allergol Int. 2022 Jan;71(1):14-24. doi: 10.1016/j.alit.2021.07.003.). Therefore, as mentioned in our paper, albeit several previous publications claim that this model is indeed useful for modeling (at least some) key aspects of not only psoriasis but also atopic dermatitis, we agree with the Reviewers and the Academic Editor that to achieve the possibly most appropriate atopic dermatitis-like cellular responses in epidermal keratinocytes, the application of certain Th2 cytokines would be desired. Therefore, we amended our manuscript and removed any reference to ‘atopic eczema’ from the text.

Minor issues:

  1. In the Introduction Section: Authors should include more appropriate references for presenting the relationship between the skin and the environment

RESPONSE:

Thank you for this important note. As requested, we included the below 4 references into the Introduction section.

  • Feingold KR, Schmuth M, Elias PM. The regulation of permeability barrier homeostasis. J Invest Dermatol. 2007 Jul;127(7):1574-6. doi: 10.1038/sj.jid.5700774. PMID: 17568800.
  • Proksch E, Brandner JM, Jensen JM. The skin: an indispensable barrier. Exp Dermatol. 2008 Dec;17(12):1063-72. doi: 10.1111/j.1600-0625.2008.00786.x. PMID: 19043850.
  • Krutmann J, Bouloc A, Sore G, Bernard BA, Passeron T. The skin aging exposome. J Dermatol Sci. 2017 Mar;85(3):152-161. doi: 10.1016/j.jdermsci.2016.09.015. Epub 2016 Sep 28.
  • Parrado C, Mercado-Saenz S, Perez-Davo A, Gilaberte Y, Gonzalez S, Juarranz A. Environmental Stressors on Skin Aging. Mechanistic Insights. Front Pharmacol. 2019 Jul 9;10:759. doi: 10.3389/fphar.2019.00759. eCollection 2019.

  1. In the Result section: The first sentence in paragraph 2.1.4 should be placed in the Introduction, while the qRT-PCR description should be moved to the Materials and Methods

RESPONSE:

Thank you for this important note. As requested, we have removed the unnecessary parts from section 2.1.4 of the Results section.

  1. In Figure 2, the reported variability is too elevated. I strongly suggest that the Authors remove this part.

RESPONSE:

Thank you for this note. We agree with the Reviewer that the variability for GM-CSF is quite high; yet, this was our experimental finding which suggests at least a tendential increase of GM-CSF upon DMG-Na treatment. However, we feel that the variability for VEGF is acceptable as the statistical analysis resulted in significantly different data. Therefore, instead of removing the complete Figure 2, we thought to keep the original Figure 2A and B (VEGF results) among the main Figure but we placed the GM-CSF (originally Figure 2C and D) results to the in the Supplementary data section as new Supplementary Figure 3C and D.

  1. In the Mat/Met Section: In paragraph 4.2 the Authors should specify the calcium concentration in the medium, strictly influencing cell differentiation.

RESPONSE:

As requested, the Ca concentration of the cell culturing medium (i.e. 264 mg/mL=1.8 mM) is now mentioned in the Materials and Methods section, 4.2. As shown in our previous publication (Papp et al, Exp Dermatol. 2003 Dec;12(6):811-24. doi: 10.1111/j.0906-6705.2003.00097.x), this medium allows the constant proliferation of pre-confluent HaCaT keratinocytes but also promotes the high-density differentiation program of the cells, initiated upon reaching confluence. As mentioned in the last sentence of 4.2, experiments of the current study were performed on pre-confluent HaCaT keratinocytes.

  1. In paragraph 4.6.1 the Authors wrote "stimulation of TLR3 activation". As it is redundant, please check the right expression.

RESPONSE:

Thank you for noting this mistake. We changed the mentioned part to ’TLR3 activation’.

  1. In the Acknowledgments: Please check

RESPONSE:

Thank you for noting this. In the original submission, which the Reviewer apparently received, we erroneously left this part unattached. After the note from the IJMS office, we corrected it with the following note: HaCaT cells were a kind gift from Prof. János Hunyadi, Department of Dermatology, University of Debrecen, Hungary.

Reviewer 2 Report

In this study, cell proliferation and anti-inflammatory effects of DMG-Na were mainly studied. However, there are some shortcomings in the research design.

1) The authors verified GM-CSF among growth factors. However, since GM-CSF often acts as an pro-inflammatory factor, it is not a good indicator for cell proliferation, and it is more appropriate to view it as an pro-inflammatory factor. Therefore, it is necessary to further verify growth factors such as FGF, TGF, and KGF rather than GM-CSF.

2) The authors used a combination of IFN and TNF as stimulators for psoriasis and atopic dermatitis. However, in the case of atopic dermatitis, IFN and TNF are Th-1 cytokines, which are cytokines that are opposed to atopic dermatitis induced by Th-2 cytokines. Therefore, a combination of Th2 cytokines such as IL-4, IL-5, IL-13, and TSLP is appropriate as a stimulator for atopic dermatitis. 

3) Although this study showed anti-inflammatory effect, the study on the molecular mechanism of DMG-Na action is missing. Inflammatory signal transduction according to stimuli, for example, NF-kB, MAPK, JAK-STAT signaling, etc., seems to require additional studies in DMG-Na treated keratinocytes activated by inflammatory stimuli.

Author Response

In this study, cell proliferation and anti-inflammatory effects of DMG-Na were mainly studied. However, there are some shortcomings in the research design.

1) The authors verified GM-CSF among growth factors. However, since GM-CSF often acts as an pro-inflammatory factor, it is not a good indicator for cell proliferation, and it is more appropriate to view it as an pro-inflammatory factor. Therefore, it is necessary to further verify growth factors such as FGF, TGF, and KGF rather than GM-CSF.

RESPONSE:

Thank you very much for this important note. We fully agree with the Reviewer that ‘GM-CSF often acts as a pro-inflammatory factor’. However, numerous studies also claim that GM-CSF indeed acts as a growth factor during e.g. skin regeneration and wound healing (see e.g. reference 54, which is now reference 72 in the revised version of the paper). Since i) we assessed the effects of DMG-Na on the synthesis and release of GM-CSF on ‘non-inflamed’, control human epidermal keratinocytes; and ii) DMG-Na exerted quite ‘universal’ anti-inflammatory effects in three different models of keratinocytes inflammation, we believe that, in this case, GM-CSF can be rightfully considered rather as a growth factor than a pro-inflammatory cytokine.

Furthermore, following the excellent idea of the Reviewer, we performed additional qRT-PCR studies on HaCaT keratinocytes treated with various concentrations of DMG-Na to measure the effects of DMG-Na on the expression of various growth factors. Indeed, qRT-PCR analysis of DMG-Na-treated HaCaT keratinocytes (2 independent experiments) revealed that 24 h DMG-Na application (up to 0.005% concentration) did not induce the mRNA level synthesis of transforming growth factor (TGF)β1 and TGFβ2 whereas the expression levels of hepatic growth factor (HGF) and epidermal growth factor (EGF) were below the detection limit both in the control and DMG-Na-treated keratinocytes, also indicating lack of induction. These new results are now shown in the new Supplementary Figure S3C (TGFβ1) and D (TGFβ2). Further, since the editorial policy of Int J Mol Sci does not allow the use of the term ‘data not shown’, our findings with HGF and EGF are only mentioned under 2.1.4.

2) The authors used a combination of IFN and TNF as stimulators for psoriasis and atopic dermatitis. However, in the case of atopic dermatitis, IFN and TNF are Th-1 cytokines, which are cytokines that are opposed to atopic dermatitis induced by Th-2 cytokines. Therefore, a combination of Th2 cytokines such as IL-4, IL-5, IL-13, and TSLP is appropriate as a stimulator for atopic dermatitis.

RESPONSE:

Thank you very much for this important note. Here, we repeat our response to the comments of the other Reviewer and the Academic Editor. Namely, as mentioned in the paper, with the inflammation experiments, our aim was to reveal the potential anti-inflammatory effects of DMG-Na in various models in which the inflammation of keratinocytes was induced by very different stimuli; in other words, we were curious to find out whether the effects of DMG-Na depends or not in the actual model and the inducer stimuli. With the joint application of INFgamma + TNFalpha, key cytokines of a plethora of inflammatory skin conditions, our goal was to induce a ‘general inflammation’ of the cells where multiple (if not all) Th-coupled inflammatory responses are engaged and augmented. We fully agree with the Reviewer (and also with the other Reviewer and the Academic Editor) that atopic dermatitis is characterized by the key involvement of Th2 cytokines and pathways. Yet, numerous prior studies also highlight that additional (Th22, Th17 and even Th1) cytokines and immune signaling mechanism also play key, pathogenic roles in the development of the disease, especially in certain subtypes (see e.g. in Brunner and Guttman-Yassky, J Allergy Clin Immunol. 2017 Apr;139(4S):S65-S76. doi: 10.1016/j.jaci.2017.01.011. and Tokura and Hayano Allergol Int. 2022 Jan;71(1):14-24. doi: 10.1016/j.alit.2021.07.003.). Therefore, as mentioned in our paper, albeit several previous publications claim that this model is indeed useful for modeling (at least some) key aspects of not only psoriasis but also atopic dermatitis, we agree with the Reviewers and the Academic Editor that to achieve the possibly most appropriate atopic dermatitis-like cellular responses in epidermal keratinocytes, the application of certain Th2 cytokines would be desired. Therefore, we amended our manuscript and removed any reference to ‘atopic eczema’ from the text.

3) Although this study showed anti-inflammatory effect, the study on the molecular mechanism of DMG-Na action is missing. Inflammatory signal transduction according to stimuli, for example, NF-kB, MAPK, JAK-STAT signaling, etc., seems to require additional studies in DMG-Na treated keratinocytes activated by inflammatory stimuli.

RESPONSE:

Thank you very much for the important note. Actually, in perfect agreement with the comment of the Reviewer, we have already started the exploration of the mechanism(s) of action of DMG-Na on the intracellular signaling pathways of the cells. Since prior publications, chiefly focusing on the modulatory effects of DMG-Na on the metabolic and oxidative status of the target cells, did not really provide guidance for specific signaling mechanisms (moreover, the biological functions described in this work have not really been described before), we established a modular strategy to obtain mechanistic data. In this strategy, we aim at assessing the effect of DMG-Na on:

  • the intracellular Ca-homeostasis of the human keratinocytes which is one of the key determinants of epidermal functions
  • the global gene-expression profile of the cells (RNASeq analysis)
  • the activity/expression of inflammatory signaling pathways and molecules – exactly those mentioned by the Reviewer.

So far, we have convincing preliminary data (obtained in 2 independent experiments) that DMG-Na induces a dose-dependent increase of intracellular Ca-concentration of human epidermal keratinocytes. However, the completion of additional mechanistic studies would require significant additional time and resources.

We truly believe that our first presentation on the versatile effects of DMG-Na on various biological functions of human epidermal keratinocytes as well as on its remarkable anti-inflammatory cutaneous actions would raise a significant interest of among the readers of Int J Mol Sci (resulting in a high number of downloads and citations) even without presenting detailed mechanistic data. Yet, to comply with the important note of the Reviewer, we included a new paragraph to the Discussion section in which i) we detail the importance of the aforementioned mechanistic studies; ii) we present the preliminary Ca-imaging data which is now shown as a new Supplementary Figure S4; and iii) discuss the potential role and impact of Ca signaling in mediating the actions of DMG-Na.

Round 2

Reviewer 1 Report

I appreciate the submitted revision and I agree with the diffusion/validity of HaCaT cells in the field of basic dermatology, but I trust that Authors tend to overestimate their work which should be better contextualized. They still include HaCaT cells among the skin models inappropriately, as the key role played by the connective tissue underlying the epidermal compartment, i.e. the dermis, in the so-called epithelial-mesenchymal crosstalk, is completely lacking. So, they should modify their approach by extrapolating the data only to the epidermis and not to the whole skin. Moreover, in the introduction they state that it "aimed at assessing the effects of N, N-Dimethylglycine sodium salt on basic biological functions and in different inflammatory in vitro models of cultured human epidermal keratinocytes", but then they present only HaCaT cells. What are the other models they refer to? If data are not presented herein, please remove this statement, otherwise misleading for the reader.

I also find an evident inconsistency between their statements on the usefulness of HaCaT cells as a "well-known, well-established, and extensively characterized cell line" and the "inherent complexity, hence heterogeneity, of any biological system"/"the actual metabolic status of the cells" to explain the large SEM values reported, as HaCaT cells are appreciated for the data reproducibility. strongly recommend that Authors must provide a convincing explanation if they want to present data in Figure 1.

Author Response

RESPONSE TO THE REVIEWER 1

I appreciate the submitted revision and I agree with the diffusion/validity of HaCaT cells in the field of basic dermatology, but I trust that Authors tend to overestimate their work which should be better contextualized. They still include HaCaT cells among the skin models inappropriately, as the key role played by the connective tissue underlying the epidermal compartment, i.e. the dermis, in the so-called epithelial-mesenchymal crosstalk, is completely lacking. So, they should modify their approach by extrapolating the data only to the epidermis and not to the whole skin.

RESPONSE:

Thank you for this important note. Albeit we had absolutely no intention to extrapolate the significance of our scientific finding to such non-epidermal functions of the skin as e.g. dermis-coupled biological responses or the mentioned epithelial-mesenchymal crosstalk, we agree with the Reviewer’s note that our scientific messages could have been more specific. Therefore, as suggested by the Reviewer, we have corrected the whole text and amended all parts in which our reference to “skin” should rather be “the epidermis” – these amendments can now be seen with the Track changes option of the Word program.

Moreover, in the introduction they state that it "aimed at assessing the effects of N, N-Dimethylglycine sodium salt on basic biological functions and in different inflammatory in vitro models of cultured human epidermal keratinocytes", but then they present only HaCaT cells. What are the other models they refer to? If data are not presented herein, please remove this statement, otherwise misleading for the reader.

RESPONSE:

Thank you for this important comment as well. Indeed, after reading the said sentence, we have to admit that it could be misleading. Therefore, we re-phrased the said sentence which now sounds: “Therefore, in this current proof-of-concept study, we aimed at assessing the effects of N,N-Dimethylglycine sodium salt on different biological functions (proliferation, migration, growth factor synthesis, inflammatory processes) of cultured human epidermal keratinocytes.”

I also find an evident inconsistency between their statements on the usefulness of HaCaT cells as a "well-known, well-established, and extensively characterized cell line" and the "inherent complexity, hence heterogeneity, of any biological system"/"the actual metabolic status of the cells" to explain the large SEM values reported, as HaCaT cells are appreciated for the data reproducibility. I strongly recommend that Authors must provide a convincing explanation if they want to present data in Figure 1.

RESPONSE:

Thank you for this valuable comment. According to our philosophy of Good Research Practice, we believe that the fundamental issue of scientific “data reproducibility”, as rightfully highlighted by the Reviewer, has two different, yet closely related, definitions hence interpretations and understandings.

The first one relates to whether independent investigators can “reproduce” the published scientific results and hence are able to draw similar conclusions by following the experimental documentation (describing e.g. the employed models, methods, read-out parameters, statistics, etc.) provided by the original investigators. With regards to the use of the human epidermal HaCaT keratinocyte cell line, we believe that the fact (which was also mentioned in our previous response to the Reviewer) that the PubMed search with the term “HaCaT” resulted in >200 publications only in Int J Derm Sci, researchers of the field do find this cell line suitable to provide reproducible data when in vitro modeling certain biological aspect of human epidermal biology. Therefore, we trust that our statement that it is a "well-known, well-established, and extensively characterized cell line" is valid.

Our interpretation of the other definition of “data reproducibility” concerns whether the inter-experimental variability of the scientific data obtained in multiple independent experiments – performed in the same model system and with the same methods and read-outs, but on different days and by a team of several investigators – is “acceptable” to draw scientifically sound and justified conclusions. As also mentioned in our previous response to the Reviewer, according to our experience with a high number of different cell lines, independent experiments performed on different days (as was the case in our studies) always results in certain variations, even when using cell lines, as part of the well-known inherent complexity, hence heterogeneity, of any biological system. Therefore, one can only hypothesize on what the true reasons are for the observed inter-experimental variability. However, we are convinced that there is another important, possibly ever more burning, question: Namely, What and who can most consistently define the degree of “acceptability” when assessing the said inter-experimental variability?

As humans tend to be subjective and hence biased, by employing the fundamental requirements of Good Research Practice, we believe that the most objective and unbiased way of defining the “acceptability” of inter-experimental variability, and then evaluating the validity of the obtained results, is to use appropriate statistical methods and analysis. As presented under “4.11. Statistical Analysis” of the manuscript, we employed a multi-level statistical analysis which is routinely used in our laboratory and which was developed by our external biostatistician expert. As all statistical methods, besides the number of independent experiments and the distribution of the data, these approaches take into account 2 major parameters when defining statistical differences: the SEM values mentioned by the Reviewer and also the mean value of the groups to be statistically compared. When interpreting the biological significance of the effect of any agent or active in a given experiment and then reliably communicating it to the scientific community, we believe that both values should be equally taken into account.

If we may, let us highlight our reasoning with 2 examples taken from the Figure 3A of the manuscript.

  1. The first example is Figure 3A which shows the qRT-PCR data obtained in the TLR3-activation induced inflammation model.
  • It is evident from measuring the mean values that both concentrations of DMG-Na exerted stronger anti-inflammatory actions in preventing the effects of the TLR3 agonist Poly-IC in upregulating the expression of IL6 than the effects of the respective concentrations of DMG-Na on IL1A. However, most probably chiefly due to the smaller inter-experimental variability (and hence the smaller SEM), the effects of DMG-Na on IL1A was statistically significant whereas on IL6 was not. So, a key question is: When interpreting the biological significance of these findings (i.e. both the degree of the anti-inflammatory effect and the results of the statistical analysis), can one rightfully claim that DMG-Na “significantly” affected both IL1A and IL6 expression?
  • Now, an “other way around” example. The SEM values for the effects of 0.0005% DMG-Na on the expressions of IL6 and IL8 are numerically almost identical (0.148 and 0.146, respectively). However, most probably chiefly due to the stronger effect of 0.0005% DMG-Na on IL8 than on IL6 expression (reflected by lower mean value), the former was statistically significant whereas the latter was not.

  1. The second example is Figure 1A which shows the results of the MTT assays.
  • At 24 hr, 0.0005% DMG-Na increased the number of viable cells by 17% when compared to the 24 hr control group. At 72 hr, however, the effect of 0.0005% DMG-Na was two times “stronger” compared to 24h as it increased the number of viable cells by 35% when compared to the 72 hr control group. So, another key question: When interpreting the biological significance of these findings (i.e. both the degree of the cell number-increasing effect and the results of the statistical analysis), which effect should one rightfully mention and highlight in the manuscript, i.e. the statistically significant but biologically rather weak (or possibly even “insignificant”) one or the statistically non-significant but biologically much stronger one?

As mentioned above, we always follow the guidelines of Good Research Practice and hence always communicate the biological significance of our findings by taking into account the results of the statistical analysis, the degree of the effect (mean values) as well as the inter-individual variability/reproducibility (SEM values). To present this philosophy, here are our answers to the above questions as well as to other parts of Figure 1, as mentioned/requested by the Reviewer:

  • Under 2.2.1, we wrote: “As assessed by qRT-PCR (Figure 3A), both tested concentrations of DMG-Na prevented the action of the inflammation-inducer TLR3-agonist polyinosinic:polycytidylic acid (Poly-IC, 20 μg/mL) to upregulate the expression of various inflammatory markers. Notably, the anti-inflammatory effects of DMG-Na were statistically significant (p<0.05, ANOVA) in the cases of genes encoding interleukin (IL)1α (IL1A), IL1β (IL1B), IL8 (IL8), and TNFα (TNFA), while the effect on IL-6 (IL6) was not significant.” Namely, albeit the statistical analysis revealed that DMG-Na statistically significantly affected 4 out of the 5 read-out inflammatory markers (i.e. all but not IL6), we believed that we rightfully included its biologically significant effect on the statistically insignificantly affected IL6 expression as well (with evidently mentioning in the referred paragraph that this was statistically “not significant”).
  • Under 2.1.1, we wrote: “DMG-Na appeared to (at least tendentially) increase the viable cell number, especially at the 72 h time points.” Namely, we used the words “appeared” and “tendentially” to not overestimate the biological significance of these findings. Further, and possibly even more importantly, we did not emphasize the statistically significant 24 hr time point data as we felt that biological significance of the small (ca. 15-17%) effects of different DMG-Na concentrations is rather vague.
  • Actually, the above MTT assay data urged us to further assess the putative cell proliferation-promoting effect of DMG-Na with 2 complementary techniques. Among these, as presented under 2.1.2 and in Figure 1D, the qRT-PCR analysis provided sound and statistically significant data on the effect of DMG-Na to upregulate the proliferation marker Ki67 gene expression. With regards to immunocytochemistry, although the % of Ki67 positive cells increased e.g. by ca. 35% (24 hr) and ca. 70% (48 hr) upon 0.005% DMG-Na administration compared to the corresponding control (Figure 1C) – an effect which, according to our vast experience, is quite remarkable when one performs immunocytochemical labeling – the statistical analysis did not result in statistically significant differences upon DMG-Na administration. However, due to the observed remarkable % change, we considered this effect as biologically significant and relevant to present and hence included these data to the manuscript. Nevertheless, in order to most reliable communicate these statistically not significant data, we used the following wording under 2.1.2: “DMG-Na increased (albeit only tendentially, due to the large standard error) the number of Ki67 positive cells after both 24 and 48 h treatment (Figure 1B, C and Supplementary Figure S1).

All in all, we are grateful to the Reviewer for his invaluable comments and strongly hope that the above reasoning is convincing to justify the presentations and interpretation of our novel scientific data.

Reviewer 2 Report

I agree to accept this manuscript in IJMS.

Author Response

RESPONSE TO THE REVIEWER 2

I agree to accept this manuscript in IJMS.

RESPONSE:

We are grateful to the Reviewer upon accepting our amendments.
